# LLM-Powered Agent for Dense Pedestrian Flow Path Planning in Train Stations

## Abstract

Managing dense pedestrian flows in train stations is a critical yet challenging task in urban transportation. Although LLM-powered agents show strong decision-making ability in many domains, they struggle in crowded, dynamic environments due to the scarcity of high-quality spatiotemporal data and effective representation methods. These limitations often cause hallucinations and poor planning. Key challenges lie in unifying macroscopic crowd metrics with microscopic individual behaviors and in efficiently encoding fine-grained spatiotemporal data under LLM context constraints. To overcome these issues, we propose **SP³Agent**, an LLM-powered agent for pedestrian flow planning that leverages a simulator, structured knowledge augmentation, and dedicated computational and analytical tools. Our simulator generates macro-scale density and velocity distributions from micro-scale trajectories, enabling holistic scene understanding. The knowledge augmentation leverages the KG-RAG framework to effectively retrieve and represent relevant spatiotemporal knowledge, while dedicated tools—such as the congestion analytics module—enable real-time, on-demand analysis of crowd dynamics. Extensive evaluations conducted in a high-fidelity environment simulated from Beijing West Station demonstrate that our agent significantly improves evacuation efficiency. Compared to conventional simulation-only crowd dynamics, our LLM agent achieves a 62% reduction in total evacuation time, a 50% decrease in average time cost, and a 21% shortening of average path length. These results demonstrate the potential of our approach in leveraging simulation-augmented data to mitigate LLM hallucination in numerical-intensive spatiotemporal decision tasks, offering a robust framework for real-world deployment in transportation hubs.

## 1 Introduction

Large train stations serve as critical hubs in urban transportation networks, handling high volumes of fluctuating pedestrian flows daily (Aiersilan, 2025). Despite their importance, managing dense pedestrian movements in such confined indoor environments remains a complex challenge (Zhang et al., 2017; Andreychuk et al., 2025). Efficient flow management is essential for optimizing system performance, enhancing passenger experience, and ensuring safety (Feng et al., 2025b). Conventional static signage or basic digital information systems often prove inadequate for managing the dynamic, high-density flows characteristic of peak hours or disruptions in such complex environments. These approaches cannot adapt in real time, predict emergent congestion, or provide personalized guidance, necessitating intelligent, adaptive planning systems (Zheng et al., 2025; Tang et al., 2024; Huang et al., 2022; Zheng et al., 2023).

An effective solution must dynamically synthesize real-time sensor data, historical flow patterns, and predictive models to generate actionable insights and personalized recommendations. As a result, optimizing internal pedestrian movement has emerged as a prominent research focus, stimulating considerable work in crowd simulation and prediction (Zhou et al., 2024; Zhang et al., 2022; Mohamed et al., 2020; Mangalam et al., 2020; Gu et al., 2022; Chen et al., 2024a; Ji et al., 2022). Although these approaches have achieved notable success, they exhibit inherent limitations. Physics-based methods overly focus on microscopic details, while data-driven deep learning approaches are often data-hungry and computationally expensive. Additionally, a lack of transparency regarding the rationale behind crowd guidance may undermine passengers' trust.

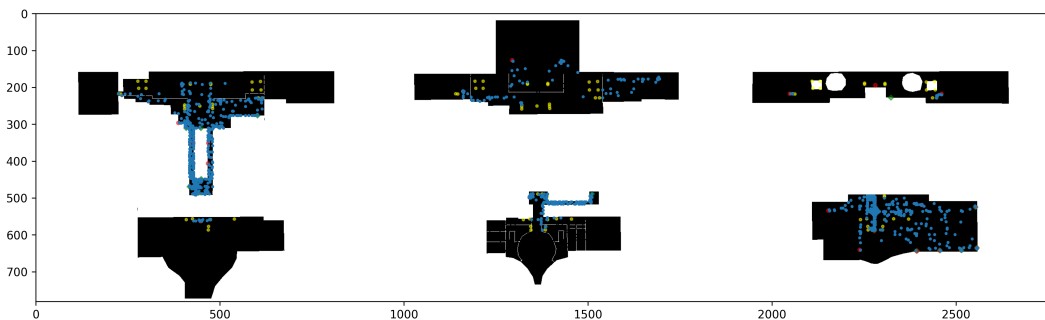

Figure 1: This visualization depicts pedestrian distribution at Beijing West Station at a given moment (with reduced traffic for clarity), showing Level -2 (left), Level -1 (center), and Level 1 (right) with their respective North/South areas; black indicates walkable zones, yellow elevators/escalators, green entrances, red exits, and blue pedestrians, revealing constrained circulation space, uneven vertical transport usage, and congestion risks in narrow areas. The 3D map is shown in Figure 5.

Recent advances in large language models (LLMs) (Yuan et al., 2025a; Guo et al., 2025; Yang et al., 2025) have enabled a new generation of AI agents capable of strong decision-making across diverse domains (Sun et al., 2023; Liu et al., 2024b; Chen et al., 2024b; Du et al., 2023; Yao et al., 2023). With their advanced capacities for semantic comprehension and reasoning, LLM-powered agents are able to address the limitations inherent in conventional simulation and prediction approaches, while also being capable of providing interpretable justifications for their decisions (Yuan et al., 2025b; Ning et al., 2025). However, their effectiveness often deteriorates in crowded, dynamic settings like train stations, where real-time adaptability and spatial reasoning are crucial for robust operation.

This paper tackles the fundamental challenge of planning dense pedestrian flows in complex train station environments. We introduce the **S**tation **P**assenger **P**ath **P**lanning Agent (**SP$^3$Agent**), a novel LLM-powered framework that leverages a simulator, structured knowledge augmentation, and dedicated analytical tools to enable intelligent, real-time crowd management. Our high-fidelity simulator is designed for predictive simulations, while the Station Knowledge Graph (StationKG) enhances the LLM's access to and reasoning about spatial semantics and facilitates relevant data retrieval. The dedicated tools compute congestion levels and provide decision recommendations by performing trajectory-based detection and AOI-guided, zone-level flow analysis. Built upon the DeepSeek-R1-0528 LLM, the agent dynamically employs external tools based on real-time pedestrian distribution and historical movement patterns to provide arriving passengers with optimized routing and destination recommendations. It further offers transparent reasoning behind its guidance to foster passenger trust. By integrating microscopic and macroscopic guidance through a coupled simulator-recommendation architecture, SP$^3$Agent effectively alleviates congestion, enhances flow efficiency, and strengthens the safety and functionality of urban transportation hubs.

The main contributions of this paper include:

- **LLM-Powered Agent:** We propose a novel LLM-powered planning agent that couples micro-level pedestrian behavior simulation with macro-level semantic reasoning. The agent analyzes both historical and real-time data, thereby enabling efficient evacuation planning and trust-aware decision-making in large-scale crowded scenarios.

- **Map and Crowd Simulator:** We introduce a high-fidelity indoor map of Beijing West Station as the spatial foundation. A novel crowd simulator, driven by a global-local potential field strategy, is then presented to enable realistic simulation and prediction of dense pedestrian flows in this complex environment.

- **StationKG and Tools:** We adopt a knowledge-driven, tool-assisted approach to enhance LLM-based spatiotemporal reasoning. This approach leverages a structured knowledge graph, constructed from simulation-generated data, to support context retrieval and spatiotemporal representation. Simultaneously, dedicated tools, such as congestion-detection modules, are applied during inference to ensure accurate and explainable decision-making.

## 2 PRELIMINARIES

### 2.1 TASK DEFINITION

Our overarching goal is to alleviate station congestion and accelerate passenger transfers via batch-level path planning. We decompose this task into (1) micro-level trajectory simulation for modeling individual movement and (2) macro-level flow regulation through batch-based destination assignments.

### 2.2 CROWD SIMULATOR

The simulator routes a crowd $\mathcal{G} = \{1, 2, \ldots, N\}$ through a station map $\mathcal{M}$ (defined in Appendix D) from their origins to destinations, adhering to natural pedestrian flow patterns. For instance, passengers who need a taxi after arriving by train start at the train exit ($o_i$) and are routed to a taxi pick-up point ($d_i$). Each individual $i \in \mathcal{G}$ has a state $\mathbf{S}_i^t = (o_i, d_i, v_i^t, \mathbf{p}_i^t)$ at time $t$, where $v_i^t \in \mathbb{R}^+$ is their velocity and $\mathbf{p}_i^t \in \mathbb{R}^2$ is their position. The simulator tracks their historical trajectory $\mathbf{T}_i^t = (\mathbf{p}_i^{0:t}, v_i^{0:t})$. The simulator's core functions are to manage the state of station resources, schedule each pedestrian's movement, and ensure all individuals eventually reach their destination ($\mathbf{p}_i^{\hat{t}} = d_i$) from their origin ($\mathbf{p}_i^0 = o_i$), as constrained by the map's topology and interpersonal interactions.

### 2.3 DESTINATION RECOMMENDATION ASSISTANT

We design an LLM-based assistant, $\mathscr{A}$, to generate destination recommendations for the newly arrived cohort at each timestep $t$. This cohort, $\mathcal{G}_t^{\text{new}} = \{i \in \mathcal{G} \mid \mathbf{p}_i^t = o_i\}$, consists of passengers who have just alighted from trains and require taxi or ride-hailing services. The assistant maps historical trajectories $\{\mathbf{T}_i^t\}_{i \in \mathcal{G}}$ and current crowd states $\{\mathbf{S}_i^t\}_{i \in \mathcal{G}}$ to recommended destinations $\{d_i \in \mathcal{D}\}_{i \in \mathcal{G}_t^{\text{new}}}$. Here, $\mathcal{D} = \{d^{(1)}, \ldots, d^{(K)}\}$ represents the candidate pick-up points. These recommendations are subject to strict proportional distribution constraints, formalized as $\frac{|\{i \in \mathcal{G}_t^{\text{new}} | d_i = d^{(k)}\}|}{|\mathcal{G}_t^{\text{new}}|} = r_k$ for all $k \in \{1, \ldots, K\}$, where $\mathbf{r} = (r_1, \ldots, r_K)$ is predefined distribution vector satisfying $\sum_{k=1}^K r_k = 1$. The assistant $\mathscr{A}$ leverages simulation-augmented data and the analytical capabilities of a reasoning LLM to enhance its decision-making. This enables holistic evaluation of real-time station conditions, allowing it to generate macro-level recommendations with explanatory insights. This approach effectively overcomes the simulator's limitations in granular, individual-level path planning.

### 2.4 OPTIMIZATION OBJECTIVE

To achieve our overarching objective, the destination recommendation assistant must generate assignments that optimize crowd distribution. Formally, this problem is framed as a constrained optimization task where the goal is to find an optimal set of destinations $d_i$ for the newly arrived cohort $\mathcal{G}_t^{\text{new}}$ that minimizes a composite cost function encompassing congestion, efficiency, and fairness.

The optimization objective is formulated as follows:

$$\min_{\{d_i\}} \mathbb{E}_{i \sim \mathcal{G}_t^{\text{new}}} \left[ \alpha \cdot \text{Congestion}(d_i) + \beta \cdot \tau_i + \gamma \cdot \text{Unfairness}(d_i) \right] \tag{1}$$

subject to the strict proportional constraints $r_k$, where (1) $\mathbb{E}i \sim \mathcal{G}_t^{\text{new}}$ denotes the expectation over all individuals in the new cohort. (2) Congestion($d_i$) quantifies the congestion impact at $d_i$ caused by assigning passenger $i$ to it. (3) $\tau_i$ represents the estimated travel time for passenger $i$ to reach their assigned destination $d_i$. (4) Unfairness($d_i$) measures the fairness cost associated with assigning destination $d_i$ to passenger $i$. (5) $\alpha, \beta, \gamma$ are predefined normalization coefficients that scale each cost term to a comparable numerical range, ensuring no single objective dominates the optimization due to differences in physical units and scales.

This formulation simultaneously addresses three critical objectives: (1) *Congestion minimization* through balanced flow allocation across available resources; (2) *Efficiency maximization* by reducing the average travel time $\mathbb{E}[\tau_i]$; and (3) *Fairness assurance* in transportation resource access. Unfairness can be measured by the variance in waiting time during each individual's journey—the smaller the variance, the higher the fairness.

# 3 METHODOLOGY

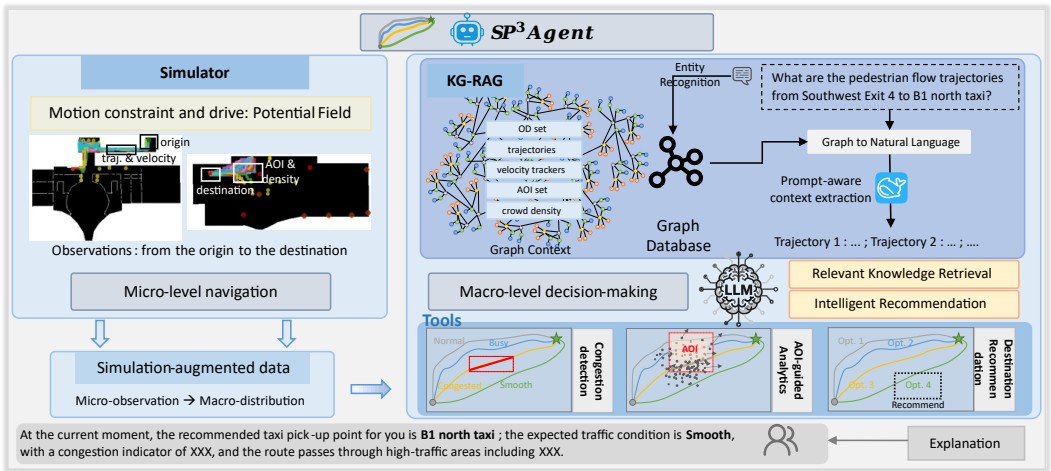

Figure 2: Overview of the SP³Agent framework.

## 3.1 OVERVIEW

We propose the Station Passenger Path Planning Agent (SP³Agent) to optimize passenger routing and alleviate congestion in high-density railway stations during peak hours. Our solution integrates two core components: a crowd simulator and an LLM-powered assistant. The simulator models micro-level pedestrian movements to generate realistic crowd trajectories, while the LLM assistant analyzes macro-level flow patterns to provide intelligent path recommendations. These modules interact in real time: the simulator supplies live crowd data to the assistant, which processes it alongside knowledge-base insights to produce decision outcomes, subsequently fed back to both the simulator and passengers. This closed-loop interaction enables dynamic and adaptive crowd management. The rest of this section is structured as follows: Section 3.2 details the crowd simulator, Section 3.3 introduces the LLM assistant. See Figure 2 for a clearer illustration of the framework.

## 3.2 POTENTIAL FIELD-DRIVEN CROWD SIMULATOR

The movement mechanism in our simulator integrates the following modules:

- Potential field global navigation: A predefined potential field is assigned to each region based on the map topology and destination. Pedestrians move along the steepest descent of the potential to follow optimal paths.

- Random walk: Stochastic movement is introduced with a certain probability to help pedestrians escape local minima and avoid deadlocks.

- Stair management: When transitioning between floors via stairs, pedestrians are held in a waiting state for a predefined duration to simulate congestion and processing time.

- Density-aware local navigation: Movement decisions incorporate local density, velocity fields, and obstacle information. Pedestrians avoid overcrowded areas and adhere to physical constraints and maximum velocity limits.

### 3.2.1 TRAIN STATION MAP

We constructed a novel, meter-scale granularity indoor map of Beijing West Station, enabling precise environmental modeling and pedestrian navigation at an unprecedented 1-meter resolution. The size of the map is $781 \times 2748$, as shown in Figure 1. The map horizontally expands the three tiers at different heights into three sections, with clear markings for passable areas, walls, doors, elevators/escalators, and key functional areas. Within our simulator, these functional areas serve as critical origin and destination points for pedestrian navigation. **Origin points** include, among others, train/subway exits and taxi/bus/ride-hailing drop-off zones, while **destination points** include

---

**Algorithm 1** Shortest Path Static Potential Field Generation

---

**Require:** $\mathcal{F}$: free space $(781 \times 2748)$, $\mathbf{s}_0$: start position, $U_0$: initial potential, $\mathcal{E}$: entry positions
1: **function** GENERATESTATICFIELD($\mathcal{F}, \mathbf{s}_0, U_0, \mathcal{E}$)
2:     $U \leftarrow \infty \cdot \mathbf{1}_{\text{size}(\mathcal{F})}$                                                      ▷ Initialize potential field
3:     $Q \leftarrow$ queue containing $(\mathbf{s}_0, U_0)$
4:     $U[\mathbf{s}_0] \leftarrow U_0$
5:     **while** $Q$ is not empty **do**
6:         $(\mathbf{p}, u) \leftarrow$ dequeue from $Q$
7:         **for** each neighbor $\mathbf{n} \in$ Neighbors($\mathbf{p}$) **do**
8:             $u_{\text{new}} \leftarrow u + 1$
9:             **if** $u_{\text{new}} < U[\mathbf{n}]$ **then**
10:                 $U[\mathbf{n}] \leftarrow u_{\text{new}}$
11:                 enqueue $(\mathbf{n}, u_{\text{new}})$ into $Q$
12:             **end if**
13:         **end for**
14:     **end while**
15:     $\mathcal{N} \leftarrow \{\mathbf{p} \mid U[\mathbf{p}] = \infty\}$                                                      ▷ Inaccessible areas
16:     $\tilde{U} \leftarrow 1 - \text{RBF}(U, U_0, 400)$                                                      ▷ Apply transformation
17:     $\tilde{U}[\mathcal{E} \cup \mathcal{N}] \leftarrow \infty$                                                      ▷ Set entries and obstacles to $\infty$
18:     $\tilde{U}[\mathbf{s}_0] \leftarrow -\infty$                                                      ▷ Set start to $-\infty$
19:     **return** $\tilde{U}$
20: **end function**

---

taxi/ride-hailing pickup points and train/subway entries, among others. This corresponds to the content in Section 2.2.

### 3.2.2 MOTION STRATEGY GUIDED BY A GLOBAL-LOCAL POTENTIAL FIELD

The simulator determines the next direction and regulates speed based on the shortest-path potential field, density and velocity potential fields. The interplay of these three types of potential fields (Wang et al., 2022), transformed via Gaussian radial basis functions, collectively drives pedestrian movement, resulting in crowd behaviors that closely approximate natural dynamics.

**Global Shortest-Path Potential field** The shortest-path potential field is computed from each map exit as the origin using the Bellman-Ford algorithm (Zhu et al., 2021) to determine the shortest distance to the exit for every accessible location within the environment. Since the spatial layout is static, the potential value associated with a given exit remains constant at each position. To optimize computational efficiency, the potential field is precomputed and stored for fast access during simulation. The pseudocode of the algorithm implementation is shown in Algorithm 1. The Radial Basis Function (RBF) kernel for mapping potential data can be written as:

$$\text{RBF}(M_p, p_{\text{exit}}, s) = \exp\left(-\frac{(M_p - p_{\text{exit}})^2}{2s^2}\right) \tag{2}$$

where $M_p$ is the shortest-path potential field matrix computed by the Bellman-Ford algorithm, $p_{\text{exit}}$ is the potential value of the station map exit, $s$ is the standard deviation.

**Local Density Potential field** The density and velocity potential fields are computed within a local window of size $(2v_{\text{max}} + 1) \times (2v_{\text{max}} + 1)$, where $v_{\text{max}}$ is the maximum pedestrian speed. The case for $v_{\text{max}} = 3$ is illustrated in Figure 6. Since the density in the immediate vicinity has a stronger influence on local congestion, we compute a weighted density potential $D_p$ by partitioning the window into $v_{\text{max}} + 1$ concentric regions based on proximity. The density potential is defined as $D_p = \sum_{i=0}^{3} w_i \cdot D_i$, where $D_i$ is the pedestrian density (persons/unit area) in the region at distance $i$ from the center, and the weights are given by $w_i = 0.1 \times (4 - i)$. The influence factor $\theta_d$ of the density potential on velocity is defined as:

$$\theta_d = 1 - \text{RBF}(D_p, D_{\text{max}}, 1) \tag{3}$$

where $D_{\text{max}} = 3$ is a maximum density parameter.

**Local Velocity Potential field** The local velocity potential $V_p$ is defined as the vector sum of all pedestrian velocities from the previous time step within a local neighborhood . It can be formulated as $V_p = \vec{v}_{\text{sum}} = \sum_{p_j \in \mathcal{N}} \vec{v}_{p_j}^{(t-1)}$, where $p_j$ denotes a pedestrian in the neighborhood $\mathcal{N}$ and $\vec{v}_{p_j}^{(t-1)}$ is its previous velocity. The influence factor $\theta_v$ of this velocity potential on the preselect velocity $\vec{v}_{\text{pre}}$ (from the shortest-path potential field) is given by:

$$\theta_v = \begin{cases} 0 & \text{if } |\mathcal{N}| = 0 \text{ or } \|\vec{v}_{\text{pre}}\|\|\vec{v}_{\text{sum}}\| = 0 \\ \frac{(1-\cos\phi)\cdot\|\vec{v}_{\text{sum}}\|}{2\cdot v_{\text{max}}\cdot|\mathcal{N}|} & \text{otherwise} \end{cases} \quad \text{where} \quad \cos\phi = \frac{\vec{v}_{\text{pre}} \cdot \vec{v}_{\text{sum}}}{\|\vec{v}_{\text{pre}}\|\|\vec{v}_{\text{sum}}\|} \quad (4)$$

where $|\mathcal{N}|$ is the number of pedestrians in the region. The resulting pedestrian velocity combines influences from all three potential fields:

$$v = \theta_d \cdot \max\left(0, \|\vec{v}_{\text{pre}}\| - \theta_v\right) \quad (5)$$

When multiple candidate velocities $\vec{v}_{\text{pre}}$ exist, the one maximizing $\cos\phi$ is selected, which aligns with the observed phenomenon of pedestrians following mainstream flow.

### 3.2.3 MOTION CONSTRAINTS

Motion constraints encompass wall obstacles, maximum area capacity, maximum pedestrian speed, random walk probability, elevator movement duration, and pedestrian obstruction tolerance. For instance, within this simulator, human prior knowledge is incorporated by setting the maximum pedestrian speed to 3 m/s, the maximum area capacity to 3 persons per square meter, the random walk probability to 10%, the inter-floor elevator travel time to 5 seconds.

### 3.3 DESTINATION RECOMMENDATION ASSISTANT

To address congestion from a system-wide perspective, we design a destination recommendation assistant that leverages LLM reasoning, knowledge augmentation, and tool-use. Sec. 3.3.1 describes the KG-RAG method for interpreting station spatiotemporal data. Sec. 3.3.2 and 3.3.3 describe two analysis tools for compute-intensive calculations and specific metric analysis, respectively.

### 3.3.1 STATIONKG: STRUCTURED STATION KNOWLEDGE ORGANIZATION

To enhance the comprehension of spatiotemporal information by LLMs in complex train station scenarios, we design a structured knowledge graph, named *StationKG*, and develop a compatible knowledge graph-based retrieval and generation framework (KG-RAG). *StationKG* is constructed from simulation-augmented data, and its schema captures the core entities and relations essential for modeling passenger mobility. The key **entities** include OD (Origin-Destination), Trajectory, SpeedTracker, AOI (Area of Interest), and DensityRecord. These entities are interconnected by fundamental **relations** such as `have`, `pass`, `share`. Our KG-RAG framework, inspired by prior work on knowledge graphs in various domains (Sanmartin, 2024; Soman et al., 2024; Wang et al., 2025; Ning et al., 2023; Liu et al., 2023; Ning & Liu, 2024), leverages the structured representations of *StationKG* to provide LLMs with the most relevant knowledge context. For detailed definitions of entities, relations, and attributes, please refer to Appendix G.

### 3.3.2 TRAJECTORY-BASED CONGESTION DETECTION

**Static Mode (S-Mode)** S-Mode computes a static congestion metric using historical trajectory data. It calculates the average path length across multiple trajectories for each origin–destination (OD) pair as an indicator of congestion. This approach is computationally efficient but relies solely on historical information, which may limit its accuracy in reflecting real-time conditions. The procedure of S-Mode is straightforward, with the congestion metric defined as:

$$C[(o, d)] = \frac{1}{|\mathcal{T}_{od}|} \sum_{T \in \mathcal{T}_{od}} |T| \quad (6)$$

where the symbols retain the same meanings as in Algorithm 2.

**Dynamic Mode (D-Mode)** D-Mode integrates both real-time and historical data to assess congestion more accurately. It retrieves historical trajectories for each OD pair, evaluates real-time pedestrian density along each trajectory, and aggregates three key metrics—average density, nonzero density ratio, and longest continuous dense segment ratio—into a weighted congestion indicator per trajectory. The final OD-level metric is the mean of these trajectory-level values. Although more computationally intensive, D-Mode offers improved real-time relevance by incorporating current crowd distribution. The algorithmic logic of D-Mode, as illustrated in Algorithm 2.

---

**Algorithm 2** OD Congestion Metric Calculation (D-Mode)

---

**Require:** OD set $\mathcal{O}$, timestep $t$, current pedestrian data $P_t$, historical pedestrian data $P_{hist}$
**Ensure:** Congestion metric dict. $C$ for each OD
 1: **for all** $(o, d) \in \mathcal{O}$ **do**
 2:     $\mathcal{T}_{od} \leftarrow$ historical trajectories for $(o, d)$ from $P_{hist}$
 3:     $S \leftarrow \emptyset$                         ▷ Set of trajectory congestion scores
 4:     **for all** $T \in \mathcal{T}_{od}$ **do**
 5:         $D_T \leftarrow \{n_x \mid x \in T, n_x = \text{count of people at } x \text{ in } P_t\}$
 6:         $\bar{d}_T \leftarrow \text{mean}(D_T)$
 7:         $r_{nz} \leftarrow |\{d \in D_T : d > 0\}|/|D_T|$
 8:         $r_{cont} \leftarrow$ longest nonzero segment length in $D_T/|D_T|$
 9:         $c_T \leftarrow 0.1 \cdot \bar{d}_T + 0.1 \cdot r_{nz} + 0.8 \cdot r_{cont}$
10:         $S \leftarrow S \cup \{c_T\}$
11:     **end for**
12:     $C[(o, d)] \leftarrow \text{mean}(S)$
13: **end for**
14: **return** $C$

---

**Comparison** S-Mode emphasizes computational speed through historical averaging, whereas D-Mode prioritizes accuracy by combining real-time density with historical patterns. The choice between modes involves a trade-off between efficiency and precision in congestion evaluation.

### 3.3.3 AOI-GUIDED PEDESTRIAN FLUX ANALYTICS

We propose an AOI-guided analysis tool to holistically evaluate congestion by combining crowd density and flow metrics. The core of this framework is a dedicated tool that computes spatio-temporal statistics for each Area of Interest (AOI). For any given AOI, our tool calculates:

- Real-time and cumulative pedestrian density, normalized by area.
- Pedestrian flux, derived from entry and exit counts per time interval.

These timestamped, AOI-indexed metrics are stored for efficient retrieval and comparative analysis, supporting both safety (density) and efficiency (flow) evaluations. This tool is integrated with a proactive analysis agent to automatically evaluate route-wide congestion through traversed AOIs, generating actionable insights for crowd management and spatial planning.

## 4 EXPERIMENTS

### 4.1 EXPERIMENT METRICS

To comprehensively evaluate the performance of our agent, we employed 16 metrics comparing scenarios before and after its adoption. The evaluated metrics include the percentage of optimized pedestrians and optimizations in average path length, average travel time, and total elapsed time.

- Path length/time optimization proportions for all/recommended group: $\rho_L^{\text{all}}, \rho_L^{\text{rec}}, \rho_T^{\text{all}}, \rho_T^{\text{rec}}$
- Average travel time and optimization indicators: $\bar{T}_{\text{base}}, \bar{T}_{\text{agent}}, \eta_{\bar{T}}, \phi_{\bar{T}}$
- Average path length and optimization indicators: $\bar{L}_{\text{base}}, \bar{L}_{\text{agent}}, \eta_{\bar{L}}, \phi_{\bar{L}}$
- Total evacuation time and optimization indicators: $T_{\text{base}}^{\text{tot}}, T_{\text{agent}}^{\text{tot}}, \eta_T^{\text{tot}}, \phi_T^{\text{tot}}$

The optimization rate ($\eta$) and improvement factor ($\phi$) for a given metric $X$ are calculated as: $\eta_X = (X_{\text{base}} - X_{\text{agent}})/X_{\text{base}} \times 100\%$ and $\phi_X = X_{\text{base}}/X_{\text{agent}}$.

## 4.2 PERFORMANCE COMPARISON ACROSS DIFFERENT CROWD SIZES

The experiments were conducted in a simulator to evaluate our agent's two modes (S-Mode and D-Mode) against a no-agent-intervention baseline across two crowd sizes (7k and 70k). The performance of several sets of comparative experiments is presented in Table 1 and 2.

Table 1: System-level performance: optimization proportions and evacuation time

| Size | Mode | $\rho_L^{\text{all}}$ | $\rho_T^{\text{all}}$ | $\rho_L^{\text{rec}}$ | $\rho_T^{\text{rec}}$ | $T_{\text{base}}^{\text{tot}}$ | $T_{\text{agent}}^{\text{tot}}$ | $\eta_T^{\text{tot}}$ | $\phi_T^{\text{tot}}$ |
|------|------|------|------|------|------|------|------|------|------|
| 7k | S-Mode | 61% | 60% | 94% | 90% | 1923 | 1649 | 14.25% | 1.17 |
| 7k | D-Mode | 55% | 57% | 55% | 59% | 1923 | 1758 | 8.58% | 1.09 |
| 70k | S-Mode | 62% | 81% | 70% | 77% | 7629 | 4229 | 44.57% | 1.80 |
| 70k | D-Mode | 57% | 77% | 51% | 67% | 7629 | 2905 | 61.92% | 2.63 |

Table 2: Individual-level performance: travel time and path efficiency

| Size | Mode | $\bar{L}_{\text{base}}$ | $\bar{L}_{\text{agent}}$ | $\eta_{\bar{L}}$ | $\phi_{\bar{L}}$ | $\bar{T}_{\text{base}}$ | $\bar{T}_{\text{agent}}$ | $\eta_{\bar{T}}$ | $\phi_{\bar{T}}$ |
|------|------|------|------|------|------|------|------|------|------|
| 7k | S-Mode | 521 | 439 | 15.76% | 1.19 | 222 | 188 | 15.30% | 1.18 |
| 7k | D-Mode | 521 | 472 | 9.41% | 1.10 | 222 | 196 | 11.32% | 1.13 |
| 70k | S-Mode | 1493 | 1184 | 20.72% | 1.26 | 1191 | 786 | 33.97% | 1.51 |
| 70k | D-Mode | 1493 | 1185 | 20.64% | 1.26 | 1191 | 592 | 50.33% | 2.01 |

Experimental results presented in Tables 1 and 2 demonstrate that our agent's two modes enhance performance against a no-agent-intervention baseline from both systemic and individual perspectives under low- and high-density conditions. Moreover, we observe that the two modes exhibit complementary strengths across different scenarios. Specifically, S-Mode leads to broader pedestrian flow optimization, achieving the best results across all performance metrics in low-density scenarios. In contrast, D-Mode excels in high-density scenarios, where its real-time computation and adaptive planning capabilities significantly reduce both individual travel times and overall evacuation duration.

## 4.3 MULTI-DIMENSIONAL CAPABILITY COMPARISON OF THE AGENT IN DENSE CROWDS

We evaluated the agent's performance in dense crowd environments across five key dimensions: evacuation efficiency (total time), temporal effectiveness (average time), congestion mitigation (waiting time), spatial utilization (AOI balance), and path optimization (average path length). The capability metrics for the scenarios without and with the agent are 1 and $1 + \eta$, respectively, where $\eta$ denotes the optimization rate. As summarized in Figure 3, both operating modes of our agent significantly outperform the no-agent-intervention baseline.

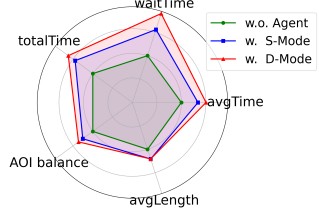

Figure 3: Capabilities Radar.

## 4.4 ANALYSIS OF PEDESTRIAN WAITING TIME AND FAIRNESS

| Wait Time | $\bar{T}_{\text{total}}$ | $\bar{T}_{\text{cont.}}^{\text{max}}$ | $T_{\text{total}}^{\text{std}}$ | $T_{\text{total}}^{\text{P95}}$ | CV | Gini |
|-----------|------|------|------|------|------|------|
| w.o. Agent | 438.04 | 283.17 | 702.01 | 1982 | 1.60 | 0.75 |
| w. S-Mode | 193.29 | 111.84 | 386.31 | 1076 | 2.00 | 0.81 |
| w. D-Mode | 43.72 | 17.57 | 119.23 | 113 | 2.73 | 0.86 |

*Note:* w./w.o. - with/without Agent; $\bar{T}$ - average time cost; cont. - continuous waiting time; P95 - the 95th percentile; CV - coefficient of variation; Gini - Gini coefficient.

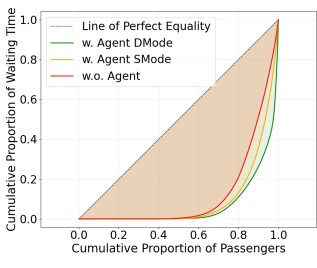

(a) Pedestrian Waiting Time  (b) Lorenz curves

Figure 4: Analysis of Pedestrian Waiting Time and Fairness.

Our analysis of pedestrian waiting time metrics (Fig. 4a) with and without our agent evaluated both congestion and fairness. The corresponding distribution is shown via Lorenz curves in Fig. 4b. Our agent, especially in D-Mode, significantly reduced most temporal metrics. While the CV and Gini coefficient worsened, we argue that absolute metrics like maximum wait time, standard deviation, and P95 are more practically relevant here. The CV and Gini coefficient can be misleading indicators of fairness under reduced congestion levels, as their values may increase even when actual pedestrian experience improves. Therefore, we prioritize standard deviation as our key fairness metric, since its interpretation of absolute dispersion better matches the practical meaning of fairness in this scenario.

## 4.5 Performance Comparison of the Agent Framework on Different LLMs

Based on the performance differences of the Agno Agent framework across various LLMs in the credibility analysis report, the results can be summarized in Table 3. The prompts for the credibility analysis and the report can be found in Appendix B and C.

Table 3: Performance Comparison of the Agno Agent Framework on Different LLMs

| Model | Reasoning | Tool-Use | Notes |
|---|---|---|---|
| QwQ-32B | - | Poor Tool-Use | Execution failed. |
| Qwen3-8B | Poor reasoning, Hallucination | ✓ | Numerical errors, logical errors |
| Qwen3-235B-A22B | Hallucination | ✓ | Limited comprehensive capability |
| DeepSeek-R1-0528 | ✓ | ✓ | Excellent reasoning |

*Note:* The "✓" symbol indicates good performance, and "—" indicates no relevant data was provided. This summary is based on limited test results with fixed prompts on specific tasks.

## 5 Related Work

**Crowd Simulation**   Crowd simulation (Zhang et al., 2017; Chen et al., 2024a; Mohamed et al., 2020; Zhang et al., 2022; Zhou et al., 2024; Aiersilan, 2025; Andreychuk et al., 2025; Pham & Bera, 2024) has seen significant advancements through the integration of physical models with machine learning. Zhang et al. (2017) introduced deep spatio-temporal residual networks for predicting city-wide crowd flows, capturing both spatial and temporal dependencies. Building on this, Chen et al. (2024a) applied a social physics-informed diffusion model to simulate crowd behaviors, enhancing the accuracy of crowd dynamics predictions. Furthermore, Zhang et al. (2022) infused physical principles into machine learning to simulate human trajectories in urban settings. In a similar vein, Zhou et al. (2024) proposed a hydrodynamics-informed neural network that models dense crowd motion, focusing on high-density crowd prediction.

**LLM-powered Intelligent Decision-making**   Recent years have witnessed a growing application of LLMs in enhancing decision-making within urban environments, particularly in transportation management and spatial optimization. For instance, in the domain of intelligent traffic systems, Ning et al. (2025) introduced Dima, an LLM-based ride-hailing assistant, while Wang et al. (2024) developed tp-GPT for traffic surveillance and control, and Lai et al. (2025) proposed Llmlight for signal control optimization. In urban spatial cognition and planning, Feng et al. (2025a) presented CityGPT for spatial management, and Li et al. (2024) designed UrbanGPT to integrate spatiotemporal information supporting dynamic urban decision-making. For path planning and prediction, Andreychuk et al. (2025) proposed Mapf-GPT for multi-agent path finding, and Liu et al. (2024a) utilized spatiotemporal LLMs for traffic forecasting.

## 6 Conclusion

In conclusion, this study presents a framework for optimizing dense pedestrian flow management in large-scale transportation hubs. The proposed LLM-powered agent combines simulation-augmented data, structured knowledge retrieval, and specialized tools to process spatiotemporal information and enhance decision-making. Utilizing DeepSeek-R1-0528 for real-time interpretation and reasoning, the system significantly improves evacuation efficiency, achieving a 62% reduction in total time, a 50% decrease in average time cost, and a 21% shorter average path length. These results demonstrate the approach's robust potential for adaptive congestion mitigation in complex station environments.

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

# A OD RECOMMENDATION BASED ON REAL-TIME DENSITY AND HISTORICAL SPEED

We develop a destination recommendation agent based on the Agno framework (AGI, 2023). The algorithm consists of four main steps: (1) identifying newly arrived passengers and their origins; (2) computing multi-dimensional congestion metrics for all relevant OD pairs; (3) ranking these pairs using a hybrid criterion that prioritizes real-time density (D-Mode) and incorporates historical average speed as a secondary weight; and (4) returning the least congested OD recommendations along with supporting parameters for dynamic routing. The detailed procedure is outlined in Algorithm 3 in Appendix A. The prompt template for credibility analysis is provided in Appendix B.

---

**Algorithm 3** OD Recommendation Based on Real-Time Density and Historical Speed

---

**Require:** Current pedestrian $P_{cur}$, sample pedestrian $P_{sample}$, timestep $t$
**Ensure:** Recommended OD set $\mathcal{R}$
 1: **Step 1:** $\mathcal{P}_t \leftarrow$ GET_PERSONS_ARRIVED_AT_TRAIN($P_{cur}, t$)
 2: $\mathcal{O}_t \leftarrow$ GET_ORIGIN_POINTS($\mathcal{P}_t$)
 3: **Step 2:** $\mathcal{C} \leftarrow$ EVALUATE_CONGESTION_METRICS($\mathcal{O}_t, t, P_{cur}, P_{sample}$)          ▷ Algorithm 2
 4: **Step 3:** $\mathcal{S} \leftarrow$ SORT_ODS_BY_DENSITY_AND_SPEED($\mathcal{C}$)
 5: **Step 4:** $\mathcal{R} \leftarrow$ SELECT_RECOMMENDED_ODS($\mathcal{S}$)
 6: **return** $\mathcal{R}$

---

# B THE CREDIBILITY ANALYSIS PROMPT TEMPLATE

**Credibility Analysis Prompt**

**Task:**
At timestep $\{timestep\}$, recommend destinations (taxi/ride-hailing pick-up points) for the passengers arriving at the train station.
**Description:**
You are an intelligent destination recommendation agent with analytical reasoning capabilities.
**Instructions:**
When a user's intention is detected as destination recommendation, invoke the tool $tool\_recommend\_dest$ with the timestep parameter $timestep = \{timestep\}$. Ensure the numerical value of the parameter is accurate!
The tool $tool\_recommend\_dest$ returns a tuple containing two elements: the first is a set of recommended origin-destination (OD) pairs, and the second is a detailed string describing the recommendation process.
Based on the returned content and data from the tool $tool\_recommend\_dest$, analyze the recommended destinations for different origins, the congestion status/level of different OD pairs, and provide reasons for the recommended OD list.
Under the sorted congestion status:

- The value after $Density$ represents the crowd congestion level calculated based on OD trajectory density (not pedestrian density).

- The value after $Velocity$ represents the crowd congestion level calculated based on historical static OD velocity trajectories from simulation data (not actual velocity).

- The value after $pathLength$ represents the average path length of the OD pair.

Higher values for $Density$ and $Velocity$ indicate higher congestion levels.
After invoking the tool, analyze the returned results before generating the output.
**Expected Output:**
# Destination Recommendations for Arriving Passengers at Timestep $\{timestep\}$:
## Recommended Destinations from Different Origins $\{recommend\_dest\_name\}$
## Congestion Status of Different Destinations $\{congestion\_state\}$
## Analysis of Recommendation Reasons $\{reason\_analyze\}$

## C  DESTINATION RECOMMENDATION ANALYSIS REPORT

---

**Destination Recommendation Analysis Report**

**Tool Calls**

- `tool_recommend_dest(timestep=996)`

**Response:**
I detected your intent for destination recommendation and will invoke the tool for detailed analysis. Below is the tool execution:

**Destination Recommendations for Arriving Passengers at Timestep 996**

**Recommended Destinations from Different Origins**

- **Origin 1:** Southwest Exit 4 entry (Coordinates: 469, 417)
    - Recommended Destination: B1_north_taxi boarding point_exit1
    - Replaces Original Destination: South Square East Boundary (Vehicle and Pedestrian Access) dbidirection
- **Origin 2:** Southeast Exit 3 entry (Coordinates: 489, 1511)
    - Recommended Destination: F1_south_taxi boarding point_exit
    - Replaces Original Destinations: B2_P3_taxi boarding point_exit and B1_north_taxi boarding point_exit2

**Congestion Status for Different Destination Routes**

| CD Path | Congestion Metric | Value | Congestion Level |
|---|---|---|---|
| SW Exit 4 → B1_north_taxi | Density | 3.52 | Very Smooth |
| | Velocity | 0.25 | |
| SE Exit 3 → F1_south_taxi | Density | 6.26 | Relatively Smooth |
| | Velocity | 0.37 | |

**Congestion Level Explanation:** 0–4 (Smooth), 5–7 (Normal), 8–9 (Busy), 10+ (Congested)

**Analysis of Recommendation Reasons**

1. **Congestion Optimization**
    - New path from Southwest Exit 4 reduces congestion by 68% (Density from 13.86 → 3.52)
    - New path from Southeast Exit 3 reduces congestion by 26–32% (Velocity from 0.54 → 0.37)
2. **Path Efficiency Improvement**
    - Average path length on Level 0: 3083 → New path: 536 (82% reduction)
    - Western path shortened by 36% (491 → 455); Southeastern path shortened by 2186% (2722 → 536)
3. **Multi-Source Integration**
    - Unified optimization for two independent passenger flows from Southeast Exit 3 (originally different destinations) to the same destination
    - Avoids intersecting bottlenecks caused by dispersed passage to multiple destinations
4. **Train Station Characteristics**
    - New destinations F1_south_taxi and B1_north_taxi are located in dedicated boarding areas within the station
    - Compared to the original destination (vehicle and pedestrian exit), the dedicated area achieves pedestrian-vehicle separation, improving safety

---

## D  DEFINITION OF ENVIRONMENT MAP

The simulator manages a railway station environment map $\mathcal{M}$ composed of accessible 1m×1m grids. To provide a detailed definition of $\mathcal{M}$, we break it into subsets that capture key functional areas within the station.

- $\mathcal{O} = \{o_1, o_2, \ldots, o_{N_o}\}$: The set of **origin points**, where $N_o$ is the number of origins. Each origin $o_k$ is a grid location where pedestrians start their journey (e.g., train exits).
- $\mathcal{D} = \{d_1, d_2, \ldots, d_{N_d}\}$: The set of **destination points**, where $N_d$ is the number of destinations. Each destination $d_k$ is a grid location where pedestrians end their journey (e.g., taxi pick-up points).
- $\mathcal{C} = \{c_1, c_2, \ldots, c_{N_c}\}$: The set of **connection points**, where $N_c$ is the number of connections. These include stairs, elevators, escalators, or other transitions that may affect pedestrian flow speed or direction.
- $\mathcal{R} = \{r_1, r_2, \ldots, r_{N_r}\}$: The set of **remaining accessible areas**, where $N_r$ is the number of such grids. These are ordinary walking areas like corridors, halls, or open spaces.

Thus, $\mathcal{M} = \mathcal{O} \cup \mathcal{D} \cup \mathcal{C} \cup \mathcal{R}$, and the subsets are mutually exclusive. Each individual $i \in \mathcal{G}$ has an origin $o_i \in \mathcal{O}$ and a destination $d_i \in \mathcal{D}$ as part of their state.

For implementation, $\mathcal{M}$ can be represented as a graph where nodes are grids and edges connect adjacent grids. Attributes such as grid type (origin, destination, connection, or rest) and walking speed modifiers (e.g., slower on stairs) are assigned. The origins and destinations are defined based on station layout:

- Origins $\mathcal{O}$: Typically train platforms, entry gates, or arrival points.
- Destinations $\mathcal{D}$: Such as taxi stands, bus stops, or exit gates.
- Connections $\mathcal{C}$: Identified from architectural plans, with dynamic properties like queueing.
- Remainder $\mathcal{R}$: Filler areas that complete the navigable space.

This detailed map ensures realistic pedestrian flow simulation by incorporating station-specific features.

## E  FIGS

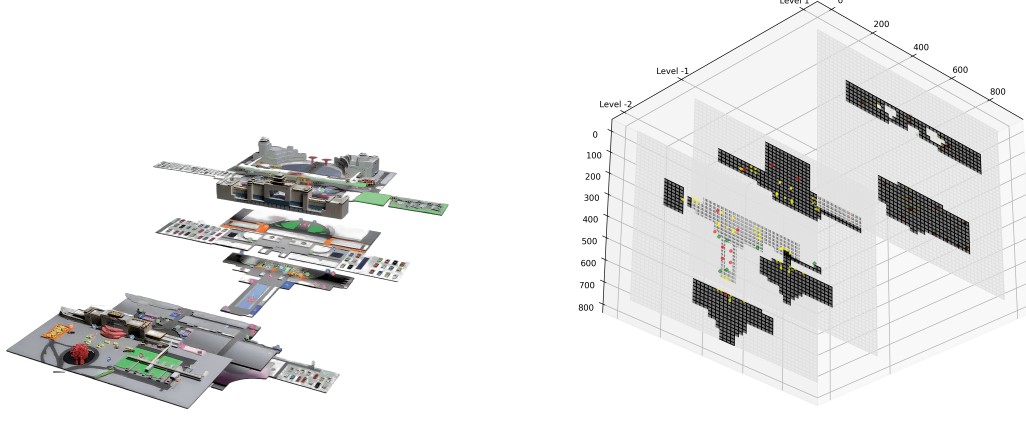

(a) Beijing West Station 3D level map              (b) Beijing West Station 3D level map simulation

Figure 5: Meter-scale indoor map of Beijing West Station for high-precision agent navigation

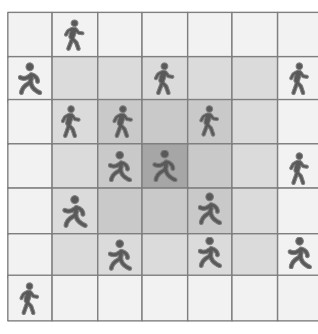

(a) Density field diagram

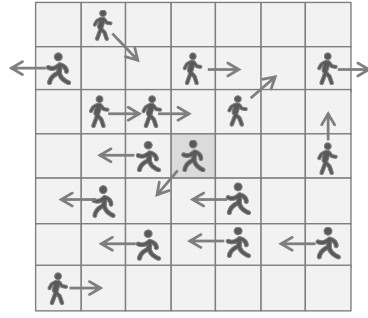

(b) Velocity field diagram

Figure 6: Schematic of potential energy calculation in density and velocity fields. Speed magnitude and direction are modulated by them, based on the shortest path potential field.

## F  D-MODE CONGESTION METRIC CALCULATION

---

**Algorithm 4** OD Trajectory Density Congestion Metric Calculation

---

**Require:** OD list $\mathcal{O}$, timestep $t$, current pedestrian $P_{cur}$, historical sample pedestrian $P_{sample}$
**Ensure:** Congestion metric dictionary $C$ for each OD
1: **for all** OD $(o, d) \in \mathcal{O}$ **do**
2:     **1. Retrieve all trajectories of the same OD from historical data**
3:     $\mathcal{T}_{od} \leftarrow$ All trajectories of $(o, d)$ in $P_{sample}$
4:     **2. For each trajectory, get the number of people at each coordinate at time** $t$
5:     **for all** trajectory $T \in \mathcal{T}_{od}$ **do**
6:         $D_T \leftarrow []$                                   ▷ Density list along the trajectory
7:         **for all** coordinate $x \in T$ **do**
8:             $n_x \leftarrow$ Number of people at coordinate $x$ in $P_{cur}$ at time $t$
9:             $D_T$.append($n_x$)
10:        **end for**
11:     **3. Calculate three metrics for the trajectory**
12:        Average density: $\bar{d}_T = \frac{1}{|D_T|} \sum_{i=1}^{|D_T|} D_T[i]$
13:        Nonzero ratio: $r_{nz,T} = \frac{|\{i \mid D_T[i] > 0\}|}{|D_T|}$
14:        Longest continuous nonzero ratio: $r_{cont,T} = \frac{\text{length of longest continuous nonzero segment}}{|D_T|}$
15:     **4. Weighted sum of the three metrics to get the congestion indicator for the trajectory**
16:        $c_T = 0.1 \cdot \bar{d}_T + 0.1 \cdot r_{nz,T} + 0.8 \cdot r_{cont,T}$
17:     **end for**
18:     **5. Calculate the mean of congestion indicators for all trajectories as the final OD congestion metric**
19:     $C[(o, d)] = \frac{1}{|\mathcal{T}_{od}|} \sum_{T \in \mathcal{T}_{od}} c_T$
20: **end for**
21: **return** $C$

---

## G  STATIONKG

As a reliable and informative knowledge base is essential for modeling and retrieving human mobility patterns within a railway station, we construct a high-quality knowledge graph called *StationKG*. We design a new schema to capture the fundamental elements of passenger movement and cover the most important relations for spatial-temporal cognition in a station environment.

In *StationKG*, the fundamental **entities** include:

- OD (Origin-Destination): The basic unit representing a movement from an origin to a destination.
- Trajectory: A specific path instance derived from an OD pair.
- SpeedTracker: A point recording speed and location at a specific time, refining a trajectory.
- AOI (Area of Interest): A functional zone within the station (e.g., ticket gate, waiting hall, platform).
- DensityRecord: A quantified crowd density value for an AOI at a specific timestamp.

Based on the types of entities, we define the typical and important **relations** to describe the connections between them as follows:

- `OD-hasTrajectory-Trajectory`
- `Trajectory-hasSpeedProfile-SpeedTracker`
- `Trajectory/SpeedTracker-passesThrough-AOI`
- `SpeedTracker-locatedIn-AOI`
- `AOI-hasDensity-DensityRecord`
- `OD-sharesOrigin-OD`
- `OD-sharesDestination-OD`

Mastering the station knowledge implies the capability of recognizing entities' important **attributes**. We select the following attributes:

- For OD: `origin, destination, timestamp_range`
- For Trajectory: `traj_id, start_time, end_time, length, avg_speed`
- For SpeedTracker: `point_id, timestamp, speed, location`
- For AOI: `AOI_id, name, type, polygon_coords`
- For DensityRecord: `record_id, timestamp, density_value`

## H    LIMITATIONS

Despite promising outcomes, certain limitations remain. First, while the simulator captures macroscopic crowd dynamics with reasonable fidelity, it still diverges discernibly from real pedestrian behavior. Validity could be improved via further training and real-world calibration to enhance generalization. Second, the current framework is limited to pedestrian flows; future work should incorporate multimodal interactions—such as vehicular flows—toward integrated human-vehicle equilibrium and system-wide efficiency gains.

## I    DISCLAIMER

The authors utilized a large language model (DeepSeek-R1) solely for the purpose of polishing and refining the linguistic expression of certain passages in this paper. All technical content, mathematical formulations, experimental results, and scientific conclusions remain the original work of the authors. The model was used as a writing assistance tool only, and all generated text was carefully reviewed, verified, and edited by the authors.

