# OpenReview forum: "LLM-Powered Agent for Dense Pedestrian Flow Path Planning in Train Stations"
_ICLR.cc/2026/Conference — Submitted to ICLR 2026_

### Official Review · Reviewer_m5mK · 2025-10-29

**Soundness:** 2
**Presentation:** 2
**Contribution:** 2
**Rating:** 2
**Confidence:** 4

**Summary:**

The paper proposes an LLM-driven agent framework (SP^3 Agent) designed to manage dense pedestrian flows in large transportation hubs. Its core idea involves generating microscopic pedestrian trajectories through a potential field-based simulator, from which macroscopic information such as density and speed distributions is extracted. Subsequently, an LLM assistant utilizes this simulation-augmented data, a knowledge graph (StationKG), and analytical tools to make macro-level decisions, recommending optimal destinations (e.g., taxi pick-up points) for newly arriving passengers.

**Strengths:**

(1) The authors propose a comprehensive, end-to-end system that integrates multiple components: a crowd simulator, a knowledge graph (StationKG), a KG-RAG framework, dedicated analytical tools, and an LLM for decision-making.

(2) The experiments are situated in a complex, large-scale environment based on a real-world location (Beijing West Station). The simulation scales (up to 70,000 pedestrians) are substantial.

**Weaknesses:**

(1) The paper's claim to "mitigate LLM hallucination" via simulation-augmented data is not adequately supported. Experimental results in Table 3 show that other LLMs (Qwen series) still exhibit severe "Hallucination" and "Numerical errors" when using the same framework. This suggests the framework's success is highly dependent on the pre-existing capabilities of a specific LLM (DeepSeek-R1-0528) rather than a generalizable feature of the proposed method.

(2) The "high-fidelity" claim appears limited to the static, meter-scale 3D map. The core crowd dynamics model relies on a classic "potential field method", which is a significant simplification of real human behavior. The paper does not quantify the gap between this model and reality, nor does it justify this choice over more modern simulators mentioned in related work. This raises serious doubts about whether the agent's learned policy is merely overfitting to the simulator's specific artifacts.

(3) The multi-objective optimization function (Equation 1) is not rigorously defined. Key terms, $Congestion(d_i)$ and $Unfairness(d_i)$, lack precise mathematical formulations. Furthermore, the paper states that the weights $\alpha, \beta, \gamma$ are "predefined" but neither specifies their values nor provides a sensitivity analysis. This renders the core objective ambiguous and the work non-reproducible.

(4) The experimental evaluation (Tables 1 & 2) is insular. It only compares the proposed S-Mode and D-Mode against a weak, internal baseline ("w.o. Agent"). The paper fails to benchmark SP^3 Agent against any state-of-the-art methods, including traditional dynamic routing algorithms or other LLM-based agents mentioned in the related work, making it impossible to assess its actual contribution.

(5) The comparison of different LLMs in Table 3 is purely qualitative (e.g., "Poor reasoning"). It lacks quantitative metrics (e.g., task success rate, numerical accuracy), key hyperparameters (e.g., temperature), and performance statistics (e.g., inference latency, token cost), which are essential for evaluating a system intended for "real-time" applications.

(6) The main experimental baseline, "w.o. Agent," is never clearly defined. It is a static, shortest-path policy based on the simulator's default potential field. This constitutes a weak "strawman" baseline. The impressive performance gains (e.g., 62% time reduction) may only show that dynamic guidance is superior to static guidance —a well-established fact —rather than demonstrating the unique merit of this specific LLM-based approach.

(7) The paper's claims regarding fairness are contradictory. While Equation 1 includes "Unfairness" as an optimization objective, the experimental results in Figure 4a show that standard fairness metrics (Gini coefficient, CV) significantly worsen under the agent's guidance. The authors' subsequent dismissal of these metrics as "misleading" is unconvincing and suggests the agent may be sacrificing system fairness for average-case efficiency, directly contradicting its stated goals.

(8) The paper does not provide sufficient evidence that a large language model is necessary for this task. Algorithm 3 and Appendix C suggest that key logic is hard-coded in the tool, and the LLM's role may be trivial (e.g., calling the tool and formatting the output). The lack of a crucial ablation study comparing the LLM agent to a simple, rule-based heuristic (e.g., "always select the tool's top-ranked option") is a major flaw.

**Questions:**

(1) Please provide a precise definition of the "w.o. Agent" baseline strategy. What logic do pedestrians follow in that condition?

(2) Please provide the exact mathematical formulas for $Congestion(d_i)$ and $Unfairness(d_i)$ as used in Equation 1. What were the values of $\alpha, \beta, \gamma$ used in the experiments? How do you reconcile the stated goal of optimizing fairness with the experimental results showing that Gini coefficient and CV worsened?

(3) Given that Algorithm 3 appears to contain the core decision logic, what specific reasoning tasks, beyond tool-calling and output formatting, does the LLM perform? Would it be possible to provide an ablation study comparing the SP^3 Agent to a heuristic algorithm that simply executes the tool's top-ranked recommendation?

(4) How can we be confident that the agent's policy is not simply overfitting to the specific dynamics and artifacts of the "potential field" simulator, and how would you expect this policy to perform given a more realistic, non-potential-field-based pedestrian model?

---

### Official Review · Reviewer_XTe9 · 2025-10-31

**Soundness:** 3
**Presentation:** 4
**Contribution:** 2
**Rating:** 4
**Confidence:** 4

**Summary:**

This paper introduces SP3Agent, an LLM-powered framework for optimizing pedestrian flow and path planning in large train stations. The proposed system integrates (1) a potential field-driven crowd simulator for micro-level trajectory modeling, (2) a knowledge-augmented LLM assistant for macro-level decision-making, and (3) dedicated analytic tools for congestion evaluation. The framework employs a structured StationKG (knowledge graph) to retrieve spatiotemporal knowledge and leverages both static (S-Mode) and dynamic (D-Mode) congestion detection modes. Experiments conducted in a high-fidelity simulation of Beijing West Station show significant improvements—up to 62% reduction in evacuation time and 21% reduction in average path length—compared with simulation-only baselines. The work argues that simulation-augmented data can mitigate LLM hallucination in numerical and spatial decision-making tasks.

**Strengths:**

Timely and relevant problem: managing pedestrian flow with LLM-based reasoning is an important and underexplored application area in urban AI and intelligent transportation.

Comprehensive system integration: successfully combines micro-level simulation, macro-level reasoning, and knowledge retrieval in a closed loop.

High-quality implementation: realistic Beijing West Station simulation, well-designed algorithms for density and velocity potential fields, and clear experimental protocols.

Significant empirical gains: strong improvements in evacuation efficiency and fairness metrics over baselines.

Excellent clarity and reproducibility, with detailed algorithm boxes and appendices.

**Weaknesses:**

Limited learning contribution: the approach does not include any learnable parameters, training process, or differentiable optimization, which weakens its relevance to ICLR’s machine learning focus.

No real-world validation: all evaluations rely on synthetic simulation; no deployment or empirical calibration is shown.

Weak baselines: comparisons are only made to simulation-only systems, without benchmarking against existing deep learning-based trajectory or crowd models (e.g., Social-STGCNN, SGAN, CrowdNet).

Ablation missing: the contribution of each component (StationKG, S-Mode vs D-Mode, LLM reasoning) is not quantitatively analyzed.

Algorithmic novelty: relies on established potential field and knowledge retrieval techniques; lacks new theoretical insight or learning formulation.

**Questions:**

Could the authors clarify whether the LLM’s reasoning is trained or purely prompt-engineered?

How does SP3Agent compare with deep-learning-based crowd forecasting models such as Social-STGCNN or diffusion-based trajectory predictors?

Can the StationKG or the congestion metrics generalize to another station layout without re-simulation?

Have the authors considered integrating reinforcement learning or policy gradient methods to make the LLM adaptive based on simulation feedback?

How robust is the system to noisy sensor input or real-world uncertainty (e.g., blocked exits, delayed flows)?

---

### Official Review · Reviewer_MBBR · 2025-11-01

**Soundness:** 2
**Presentation:** 2
**Contribution:** 2
**Rating:** 2
**Confidence:** 4

**Summary:**

This paper addresses a destination recommendation problem, where crowd that just arrived at a station are assigned to one of the destinations that are taxi pick up points. The authors developed two modules:
- crowd simulator
- LLM-based destination recommender.

**Strengths:**

- Relatively new use of LLMs: Make them solve a classical OR problem.
- Implementation of crowd simulator with a customized potential model.

**Weaknesses:**

The problem looks like the same as resource allocation problem or vehicle routing problem. The paper doesn't compare the proposed method with such well-known solutions. Additionally, quite limited information is provided how LLMs are used to solve the optimization problem. . This is problematic, given that LLM might not be the best model to solve routing problems, according to [Chen+24].

@article{chen2024can,
  title={Can LLMs plan paths in the real world?},
  author={Chen, Wanyi and Su, Meng-Wen and Mehjabin, Nafisa and Cummings, Mary L},
  journal={arXiv preprint arXiv:2411.17912},
  year={2024}
}

As a result, it is not clear that what technical contributions this papers made. I don't support acceptance of the paper in the present form.

**Questions:**

- Section 2.2 says that the simulator "tracks" historical trajectories. Does it follow the previously observed trajectories, or does it randomly generate new trajectories that may not be in the dataset?


- How does LLM solves the optimization problem? Why does the standard MILP formulation is not an option?

---

### Official Review · Reviewer_z81n · 2025-11-02

**Soundness:** 3
**Presentation:** 3
**Contribution:** 3
**Rating:** 6
**Confidence:** 1

**Summary:**

The paper proposes SP3Agent, a novel agent designed to optimize pedestrian flow management in crowded train stations using large language models. SP3Agent combines a high-fidelity crowd simulator, structured knowledge augmentation, and specialized analytical tools to overcome challenges in pedestrian movement planning, such as crowd congestion and inefficient pathfinding. By leveraging simulation-augmented data, the agent improves evacuation efficiency by significantly reducing total evacuation time, average travel time, and path length compared to conventional methods. The system demonstrates its effectiveness through extensive evaluations in a high-fidelity environment based on Beijing West Station, offering a robust framework for real-world applications in transportation hubs.

**Strengths:**

1. The paper introduces a unique approach by combining LLM-powered decision-making with a crowd simulation framework, enabling both micro-level pedestrian behavior simulation and macro-level flow regulation in real-time, which has not been widely explored in this context.
2. The paper presents a well-detailed methodology, including novel simulation techniques, a custom knowledge graph (StationKG), and analytical tools like congestion detection modules. The experimental design and results offer strong evidence of the system's efficacy in dense crowd settings.
3. The paper is structured clearly, with well-explained figures and algorithms that demonstrate the inner workings of SP3Agent and its evaluation. Complex concepts like potential fields and congestion metrics are explained in an accessible way.
4. The proposed framework addresses a critical challenge in urban transportation systems, with significant potential for improving pedestrian flow and safety in crowded environments, which can directly impact public transportation efficiency and safety.

**Weaknesses:**

1. Simulation Fidelity: Although the simulator is high-fidelity, the authors acknowledge that it still diverges from real pedestrian behavior. Future improvements in calibration using real-world data could enhance its accuracy.
2. Model Complexity: The system's reliance on complex algorithms and multiple tools may increase computational costs. For real-time deployment in large-scale stations, optimizing the computational efficiency of the system could be essential.

**Questions:**

1. How does SP3Agent handle unexpected disruptions or changes in pedestrian behavior, such as sudden crowd surges or blockages? Is the system capable of adapting to these unforeseen changes in real-time?
2. Given the current limitations of the simulator, how might you improve its calibration with real-world data?

---

### Meta-Review · Area_Chair_JDmc · 2026-01-12

**Summary:**

The paper proposes SP3Agent, an agent-based framework that leverages large language models to optimize pedestrian flow management in crowded train stations. While the work presents an interesting and novel application of LLMs to a classical operations research problem, several critical concerns remain. In particular, the experimental evaluation is limited to simulation-based comparisons, and it is unclear how the proposed approach is calibrated or validated against real-world pedestrian flow data, which raises questions about its practical applicability and robustness.

**Reviewer Concerns:**

No rebuttal was submitted.

**Reviewer Scores:**

No rebuttal was submitted.

---

### Decision · Program_Chairs · 2026-01-26

Reject